# JARA: Joint Alignment and Reconstruction Architecture for Region-Aware Vision-Language Pretraining

## Abstract

Contrastive Language-Image Pretraining (CLIP) shows strong zero-shot transfer capabilities. However, it fails to capture the intrinsic semantic structure within images and performs weak on fine-grained retrieval and dense prediction. In this work, we propose Joint Alignment and Reconstruction Architecture (JARA), a unified framework that integrates region-aware learning into CLIP via self-supervised objectives. JARA employs a Spatially Balanced Masking (SBM) strategy to decouple each image into context and masked regions uniformly. On this basis, JARA firstly replaces vision-to-vision self-distillation with Cross-Modal Self-Distillation (CMSD) to align context region's `[CLS]` tokens with paired captions. Secondly, JARA extends multi-view learning to semantic patch reconstruction to encourage the model to learn the intrinsic association across image regions, enabling region-level semantics to synchronously emerge during contrastive training. Both objectives are optimized in the same masked view, achiving an efficient single-pass training. Experiments on image-text retrieval and open-vocabulary segmentation show that JARA achieves state-of-the-art performance while remaining efficient. The code will be available after the review phase.

## 1 Introduction

Recent advances in Vision-Language Pretraining (VLP), such as CLIP (Radford et al., 2021) and ALIGN (Jia et al., 2021), have shown strong zero-shot and open-vocabulary recognition ability using large-scale image-text contrastive pairs. However, their performance on fine-grained retrieval and dense prediction tasks (*e.g.*, segmentation) remains limited. As shown in Fig. 1b, CLIP's feature maps are plagued by noise and might miss visual components in the images. A key reason is that current VLP pipelines typically learn global image-level representations and lack patch-level representations modeling. Moreover, web-crawled captions for CLIP provide weak and incomplete descriptions of image contents. In this case, CLIP facilitates only coarse feature learning for fine-grained image representations and offers insufficient alignment for dense tasks. To address this, one possible solution is to collect massive region captions or detection annotations, and then enforce region-text alignment for better dense feature semantics, like RegionCLIP (Zhong et al., 2022), UMG-CLIP (Shi et al., 2024), and FG-CLIP (Xie et al., 2025). Yet such approaches require costly annotations, heavy data cleaning, and often rely on private resources, which limits reproducibility.

An alternative is to integrate Self-Supervised Learning (SSL) into CLIP. Specifically, SSL provides richer multi-view image–text pairs (Li et al., 2021), while masked prediction, by reconstructing the masked regions, drives the model to capture intra-image structures and thus produce semantically coherent dense features across image regions (Zhou et al., 2021; He et al., 2022; Oquab et al., 2024). For example, CLIPSelf (Wu et al., 2024), SILC (Naeem et al., 2024) and FineCLIP (Jing et al., 2024) adopt multi-view Self-Distillation (SD) *(also called Dino-style SD)* to enforce consistency between global and local crops for fine-grained image representations. Moreover, EVA-CLIP (Sun et al., 2023) and TIPS (Maninis et al., 2025) adopt token-level Masked Image Modeling (MIM) *(also called iBoT-style MIM)* to model dense representation for image region. However, Dino-style SD and iBoT-style MIM both were originally designed for vision tasks. Simply stitching them onto CLIP will inevitably lead to the following issues:

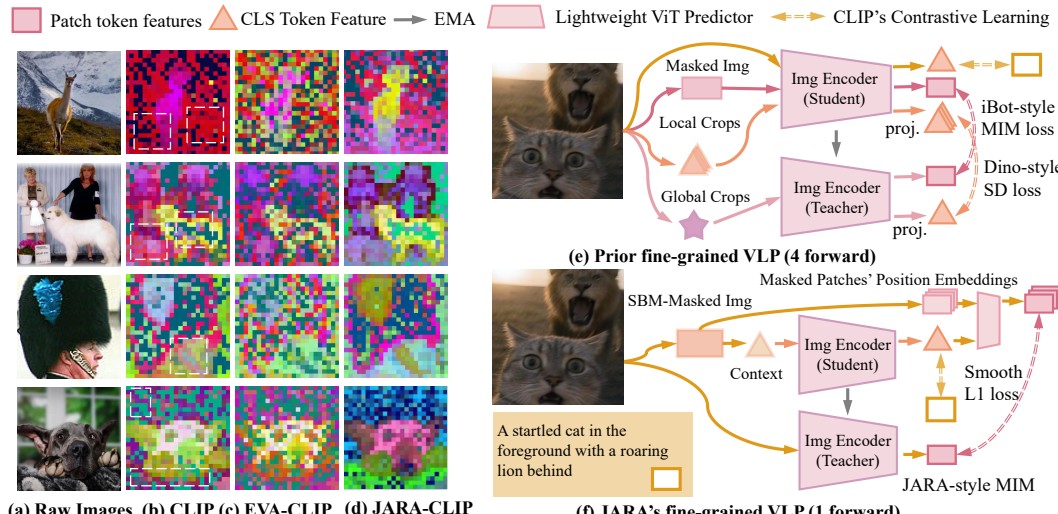

Figure 1: (a)-(d) Illustration of different VLP methods' dense feature maps. RGB channels represent the top three principal components. (e)-(f) Comparison of different VLP methods. The "**Context**" in (f) indicates the unmasked patches. Different color lines indicate different forward processes.

*(i) Isolation bias issue.* They often lead to visual features being learned in isolation from text supervision, limiting cross-modal alignment.

*(ii) Contrastive disruption issue.* Masking introduces additional noise that disrupts image-text contrastive learning. Although existing methods (Naeem et al., 2024; Jing et al., 2024; Sun et al., 2023; Maninis et al., 2025) attempt to address this by performing multiple training stages or forward passes (*e.g.*, masked, full, and multi-view images, marked in different colors in Fig. 1e), this solution substantially reduces training efficiency.

To address the aforementioned issues, we propose the **Joint Alignment and Reconstruction Architecture (JARA)**, a unified framework that integrates SSL directly within CLIP image-text contrastive alignment without additional region captions and locating annotations. The overall pipeline can be found in Fig. 1f. **Firstly**, we decouple the input image into one visible context region and multiple masked regions. Beyond DINO-style SD in SILC (Naeem et al., 2024) and FineCLIP (Jing et al., 2024), we unify SD with CLIP's image-text contrastive learning into a Cross-Modal Self-Distillation (CMSD) process, aligning the context `[CLS]` with image's caption. This allows local views to learn global semantics directly from text describing the full image, rather than from a larger visual view, thereby strengthening region-aware alignment. **Secondly**, we extend the idea of multi-view distillation to patch-level learning. Inspired by JEPA (Assran et al., 2023), masked patch embeddings are predicted from the visible context via a lightweight predictor, supervised by a teacher encoder processing the full image (*namely JARA-style MIM, see magenta dashed lines in Fig. 1f*). This encourages each visible region to encode contextual cues useful for reconstructing its masked counterparts, yielding semantically coherent dense representations. Unlike EVA-CLIP (Sun et al., 2023) and TIPS (Maninis et al., 2025), which reconstructs masked embeddings directly from pixels and risks introducing noise to image-text alignment, JARA ensures both efficiency and cross-modal consistency. **Finally**, we observe that random masking in DINO/iBOT (Zhang et al., 2022; Oquab et al., 2023; Zhou et al., 2021) often oversamples central regions and causes spatial bias *(see Fig. 3a)*. By design, we propose **Spatially Balanced Masking (SBM)** to ensure more uniform spatial coverage. It enables more consistent feature learning across central and peripheral regions, yielding higher-quality dense representations.

At a high level, the solution of JARA resembles solving a jigsaw puzzle with missing pieces: if the observed patches are disordered or ambiguous, it becomes nearly impossible to infer the full image. To succeed, the model must ensure that each visible patch is semantically coherent and informative enough to support holistic reconstruction. This helps JARA uncover the intrinsic semantic structure within existing image-text pairs, enabling region-level semantics to emerge naturally alongside image-text alignment. Our contributions are summarized as follows:

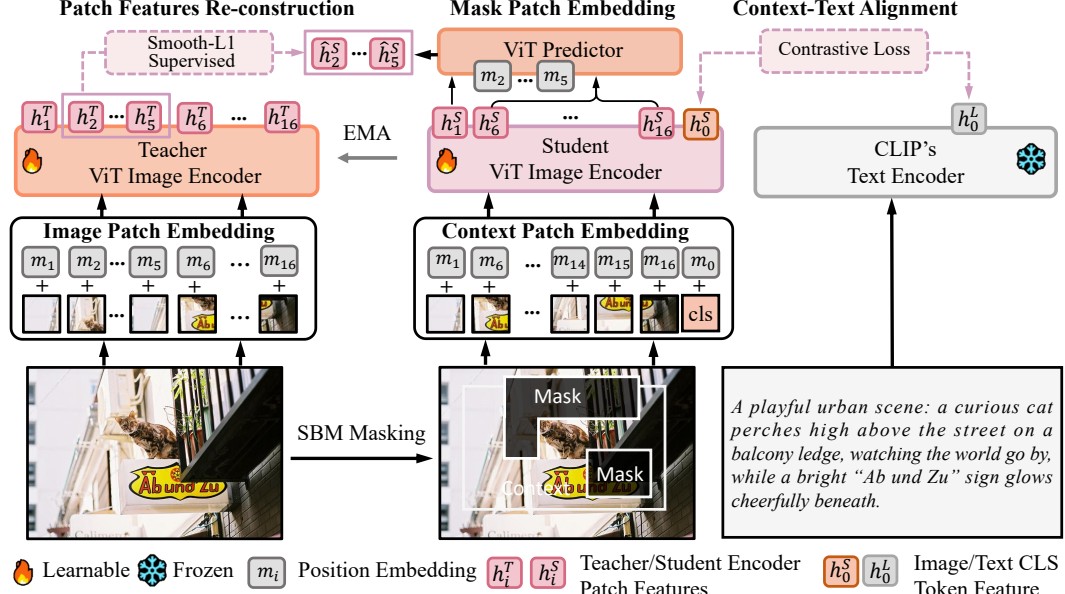

Figure 2: Overview of the JARA framework. We decouple the CLIP image encoder input into a visible *context* region and multiple *masked* regions by a Spatially Balanced Masking (SBM) strategy. The context patch goes into the student encoder, and the masked patches go to a predictor, avoiding conflicts between image-text alignment and reconstruction. All objectives are unified within a single forward pass.

- By leverging CMSD and JARA-style MIM, JARA is the first to unify cross-modal alignment and semantic reconstruction within a single-pass framework. JARA enables efficient and region-aware learning without any region-level annotations or synthetic supervision..

- We identify the spatial bias in DINO's masking strategy and introduce Spatially Balanced Masking (SBM) to achieve a uniform random masking.

- Without relying on curated or annotated region-level data, we benchmark JARA solely on vanilla open-source VLP data and demonstrate state-of-the-art performance in image-text retrieval and open-vocabulary segmentation.

## 2 RELATED WORK

**Learning fine-grained representation with detailed caption or region locating data.** Contrastive VLP relies heavily on large-scale, high-quality image-text pairs. OpenAI CLIP collected 400M private WebImageText pairs (WIT-400M) (Radford et al., 2021), while its cleaning pipeline is non-public. For its reproducible version, *i.e.*, OpenCLIP (Cherti et al., 2023), it was trained on 2B (5×400M) noisy web-crawled image text pairs (Laion2B) to achieve comparable performance, showing that the quality of VLP is largely shaped by data scale, type, and quality (Chen et al., 2024). To achieve a fine-grained representation, several methods (Zhong et al., 2022; Jing et al., 2024; Varma et al., 2023; Kim et al., 2024) introduce region-text alignment based on synthetic or weak supervision. RegionCLIP (Zhong et al., 2022) introduces 3M region captions from CC3M to align patch embeddings with textual regions. FG-CLIP (Xie et al., 2025) re-captions Laion2B using large language models (*e.g.*, CogVLM2-19B), and employs open-vocabulary detection models (*e.g.*, YOLO-World) to ground concepts before producing fine-grained region captions. In terms of fine-grained representation at the image level, ShareGPT4V (Chen et al., 2024) provides GPT4-Vision-curated captions that support subsequent methods such as LongCLIP (Zhang et al., 2024), TULIP (Najdenkoska et al., 2024), and RetainCLIP (Feng et al., 2025). In JARA, instead of focusing on data, we exploit the intrinsic semantic structure of images to enhance CLIP's fine-grained representations at both global and regional levels.

**Learning fine-grained representation with SSL.** Self-supervised objectives are widely used to strengthen fine-grained visual features. Broadly, these methods can be grouped into multi-view self-distillation and masked image modeling.

Multi-view self-distillation, (*e.g.*, DINO (Caron et al., 2021), DINOv2 (Oquab et al., 2023)), enforces consistency between global and local crops via a teacher–student setup, encouraging an image-level [CLS] representation that captures holistic semantics from partial observations (See orange and pink lines of Fig 1e). SILC (Naeem et al., 2024) integrates this strategy into CLIP, aligning local–global views to enhance fine-grained retrieval. CLIPSelf (Wu et al., 2024) further distills dense region features to their corresponding image-crop embeddings, improving open-vocabulary dense prediction. FineCLIP (Jing et al., 2024), DeCLIP (Wang et al., 2025) further extend this idea by incorporating additional teacher. Instead, masked image modeling reconstructs masked content from visible context to learn patch-level representations (See magenta and pink lines of Fig 1e). Pixel-level methods, (*e.g.*, MAE (He et al., 2022), SimMIM (Xie et al., 2022)), often bias toward low-level textures, while token/latent-level methods, (*e.g.*, BEiT (Bao et al., 2021), iBOT (Zhou et al., 2021), JEPA (Assran et al., 2023)), emphasize higher-level semantics. In VLP, *EVA-CLIP* (Sun et al., 2023) injects pixel-level reconstruction into CLIP training, while FineCLIP (Jing et al., 2024) and TIPS (Maninis et al., 2025) adopt iBOT-style masking, with TIPS further combining DINOv2 losses to strengthen patch features. SF-CLIP (Sameni et al., 2024) trains CLIP with optionally masked image and corresponding text, and conducts distillation from large foundation models, guiding more robust compositional features. Some more previous methods also try to combine contrastive-based SSL methods with CLIP such as SimCLR-CLIP (Mu et al., 2022).

Although effective, most of these methods are direct transplants from vision-only SSL. In contrast, JARA unifies these objectives within image–text alignment in a single-pass framework, avoiding such drawbacks.

## 3 METHOD

### 3.1 PRELIMINARIES

**CLIP** (Radford et al., 2021) aligns image and text embeddings within a shared latent space by contrastive learning. Given a batch of image-text pairs $\{(x_i, t_i)\}_{i=1}^B$, the image encoder $f_I$ and the text encoder $f_L$ with [CLS] features $h_i^I = [f_I(x_i)]_{\text{cls}}$ and $h_i^L = [f_L(t_i)]_{\text{cls}}$, respectively. An InfoNCE-style (Oord et al., 2018) symmetric contrastive objective is used, composed of an image-to-text (I2T) loss $\mathcal{L}_{\text{I2T}}$ and a text-to-image (T2I) loss $\mathcal{L}_{\text{T2I}}$, and the final loss is their simple average.

**Joint Embedding Predictive Architecture (JEPA)** is a SSL method that learns semantic structure by predicting masked patch features in the latent space. Given an input image divided into $N$ patches, a subset $\mathcal{M} \subset \{1, \ldots, N\}$ is randomly picked as *Mask*, and the remaining visible set $\mathcal{V} = \{1, \ldots, N\} \setminus \mathcal{M}$ is treated as the *Context*. The student encoder $f_I^S$ processes the context patches $\{x_j\}_{j \in \mathcal{V}}$, and a predictor estimates the semantic embeddings $\{\hat{h}_k^S\}_{k \in \mathcal{M}}$ for the masked patches. The ground-truth targets $\{h_k^T\}_{k \in \mathcal{M}}$ are provided by a teacher encoder $f_I^T$, whose parameters are updated as the exponential moving average (EMA) (Tarvainen & Valpola, 2017) of the student encoder $f_I^S$. Training minimizes a Smooth-L1 loss between $\hat{h}_k^S$ and $h_k^T$.

### 3.2 APPROACH

**Overview.** JARA enhances vision-language pretraining by jointly optimizing two complementary objectives:

$$\mathcal{L} = \mathcal{L}_{\text{CL}} + \lambda \cdot \mathcal{L}_{\text{REC}}. \tag{1}$$

Here, $\mathcal{L}_{\text{CL}}$ denotes the contrastive alignment loss, $\mathcal{L}_{\text{REC}}$ the reconstruction loss, and $\lambda$ a hyperparameter. The pipeline is illustrated in Fig. 2. A key property of JARA is that both objectives operate on a single masked view, ensuring efficiency and avoiding conflicts.

**Cross-Modal Self-Distillation (CMSD).** We apply our Spatially Balanced Masking (SBM) to divide an image $x_i$ into a visible subset $\mathcal{V}$ and a masked subset $\mathcal{M}$:

$$x_i \longrightarrow \{x_i^{\mathcal{V}}, x_i^{\mathcal{M}}\}. \tag{2}$$

The student encoder $f_I^S$ processes only $x_i^{\mathcal{V}}$, producing a context `[CLS]` token $h_{\text{cls},i}^S$. Unlike DINO-style vision-to-vision self-distillation, JARA aligns $h_{\text{cls},i}^S$ with the `[CLS]` of its paired caption $h_{\text{cls},i}^L$ (from frozen text encoder $f_L$), unifying self-distillation and CLIP-style contrastive learning into a single cross-modal objective:

$$\mathcal{L}_{\text{I2T}} = -\frac{1}{N}\sum_{i=1}^N \log \frac{\exp(\cos(h_{\text{cls},i}^S, h_{\text{cls},i}^L)/\tau)}{\sum_{j=1}^N \exp(\cos(h_{\text{cls},i}^S, h_{\text{cls},j}^L)/\tau)}, \mathcal{L}_{\text{T2I}} = -\frac{1}{N}\sum_{i=1}^N \log \frac{\exp(\cos(h_{\text{cls},i}^L, h_{\text{cls},i}^S)/\tau)}{\sum_{j=1}^N \exp(\cos(h_{\text{cls},i}^L, h_{\text{cls},j}^S)/\tau)}.$$

where $h_{\text{cls},i}^L$ is the text embedding for caption $t_i$, and $\tau$ is the temperature. To handle noisy captions that describe only part of the image (*partial labeling*), we follow the asymmetric contrastive loss (Yao et al., 2023) to downweight $\mathcal{L}_{\text{I2T}}$ and emphasizes $\mathcal{L}_{\text{T2I}}$. This reduces the effect of unmatched regions and yields more quality alignment. CMSD allows local views to learn global semantics directly from text describing the full image, rather than from a larger visual view, thereby sovling the *isolation bias issue*. And at the same time, the image-text alignment is achieved.

**JARA-style Masked Image Modeling (JARA-style MIM).** To further model intra-image structure, JARA incorporates a JEPA-style semantic reconstruction branch. The student encoder $f_I^S$ outputs the embeddings of the context patches, denoted as $H_i^{\mathcal{V}} = \{h_{k,i}^S\}_{v_k \in \mathcal{V}}$. For each masked position $k \in \mathcal{M}$, we introduce a learnable mask embedding $n_k$ combined with its sinusoidal positional embedding $p_k$, written as:

$$z_k = n_k + p_k. \tag{3}$$

The set of all such mask embeddings for the image $x_i$ is denoted as $Z_i^{\mathcal{M}} = \{z_{k,i}\}_{k \in \mathcal{M}}$. Then a lightweight ViT predictor estimates the semantic embeddings of masked patches by concatenating both the context embeddings and the mask embeddings as input:

$$\hat{H}_i^{\mathcal{M}} = \text{ViT-Predictor}\big([H_i^{\mathcal{V}}; Z_i^{\mathcal{M}}]\big). \tag{4}$$

A teacher encoder $f_I^T$, updated via EMA, processes the full image $x_i$ and produces target embeddings $H_i^{\mathcal{M}}$ for the masked patches' embedding. The reconstruction loss is computed as a Smooth L1 loss between $\hat{H}_i^{\mathcal{M}}$ and the teacher targets $H_i^{\mathcal{M}}$, encouraging context patches to encode predictive and semantically rich features. By conditioning masked patches on context embeddings, the predictor enables masked feature reconstruction and image-text alignment within a single forward pass. Combined with CMSD, this addresses the *contrastive disruption issue* without extra stages or passes.

### 3.3 SPATIALLY BALANCED MASKING (SBM)

Existing masking strategies in iBoT/DINO/DINOv2 (Zhou et al., 2021; Caron et al., 2021; Oquab et al., 2024) rely on random rectangular cropping. Since sampled boxes must remain inside image boundaries, many edge candidates are discarded, causing central regions to be over-sampled while peripheral patches are rarely masked. This spatial bias reduces coverage diversity and limits region-level representation learning. To address these, we propose SBM as follows.

**Patch Frequency–Aware Sampling.** We track the historical masking count $F_i$ of each patch $i$ and assign sampling weights inversely related to $F_i$:

$$w_i = \frac{1}{1 + \exp(F_i)} \Big/ \sum_j \frac{1}{1 + \exp(F_j)}. \tag{5}$$

This favors rarely masked patches and achieves more uniform spatial coverage over training.

**Offset-Aware Region Sampling.** To avoid repeatedly masking local clusters, we add an offset $(\delta_x, \delta_y)$ to the candidate window center. The offset range expands with the neighborhood's average masking frequency $\bar{F}_n$, pushing masks outward in heavily sampled areas:

$$\delta_x \sim \text{Uniform}\big(-\tfrac{h}{2}(1 + \bar{F}_n), \tfrac{h}{2}(1 + \bar{F}_n)\big), \quad \delta_y \sim \text{Uniform}\big(-\tfrac{w}{2}(1 + \bar{F}_n), \tfrac{w}{2}(1 + \bar{F}_n)\big). \tag{6}$$

SBM balances central and peripheral sampling, mitigates the bias of random masking, and yields more consistent region-aware features and higher-quality dense representations (see Fig. 3ab). Pseudocode and more details for SBM is in Appendix C.

Table 2: Ablation on masked image modeling design and validation on ZSSeg task.

(a) **Ablation on training strategies.**

| Full Image -Text Contrast | Context Image -Text Contrast | JARA-Style MIM | Flickr30K-test I2T R@1 | T2I R@1 |
|---|---|---|---|---|
| ✓ | ✗ | ✗ | 87.1 | 66.2 |
| ✗ | ✓ | ✗ | 87.2 | 68.2 |
| ✗ | ✓ | ✓ | **87.5** | **74.5** |

(b) **ZSSeg Validation on COCO.**

| Model | Backbone | aAcc | mIoU | mAcc | mPrec |
|---|---|---|---|---|---|
| CLIP | ViT-B/16 | 23.7 | 12.5 | 27.1 | 23.1 |
| JARA-CLIP | ViT-B/16 | **26.0** | **13.1** | **27.3** | **24.5** |
| CLIP | ViT-L/14 | 11.1 | 8.3 | 14.8 | 25.7 |
| JARA-CLIP | ViT-L/14 | **19.1** | **10.6** | **19.3** | **28.6** |

# 4 EXPERIMENTS

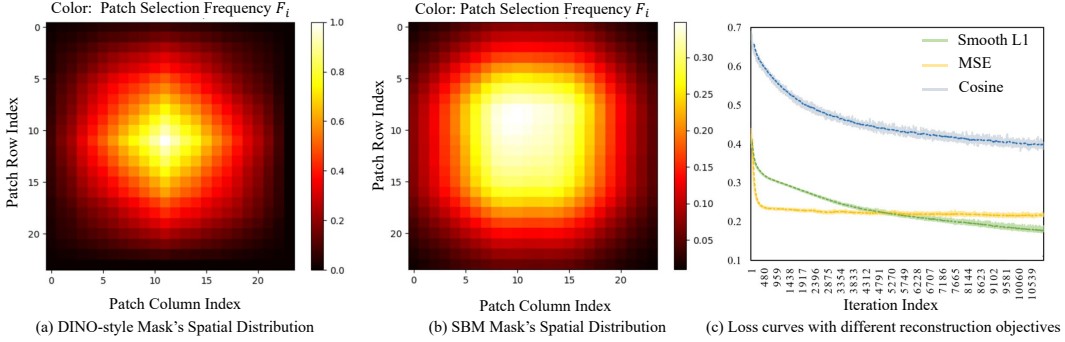

(a) DINO-style Mask's Spatial Distribution  (b) SBM Mask's Spatial Distribution  (c) Loss curves with different reconstruction objectives

Figure 3: Illustration of the spatial distribution of different mask strategies and the loss curves with different reconstruction objectives.

Table 1: Ablation on JARA's architecture design.

(a) **Different mask strategies.**

| Mask Type | COCO I2T R@1 | T2I R@1 | ADE-847 mIoU | ADE-150 mIoU |
|---|---|---|---|---|
| CLIP (w/o mask) | 59.4 | 38.9 | 8.4 | 27.2 |
| + REC w. Uniform Mask (Li et al., 2022) | 57.3 | 40.5 | 12.1 | 32.3 |
| + REC w. DINO-style Mask | 58.1 | 44.7 | 12.1 | 32.1 |
| + REC w. **SBM** Mask | **62.6** | **49.0** | **12.8** | **33.0** |

(b) **Different reconstruction losses.**

| Loss Type | COCO I2T R@1 | T2I R@1 |
|---|---|---|
| MSE | 60.1 | 47.4 |
| Cosine | 54.9 | 39.5 |
| **Smooth L1** | **62.6** | **49.0** |

**Implementation Details.** The model is initialized with weights from OpenAI CLIP (Radford et al., 2021). For ViT-L/14 and ViT-B/16 (Dosovitskiy et al., 2020), the batch sizes per Ascend 910B NPU are set to 80 and 448, respectively. We adopt the AdamW optimizer with a learning rate of $1 \times 10^{-5}$, weight decay of 0.05, $\beta_1 = 0.9$, and $\beta_2 = 0.98$. A linear warmup followed by a cosine learning rate schedule is applied. The weight for the reconstruction (REC) loss:contrastive learning (CL) loss set to $\lambda = 2$. More details (hidden dim, bs, *etc., al.*) can be found in Appendix A.

**Fair Comparison.** As discussed in Sec. 2, CLIP's performance is highly sensitive to data quality, particularly whether the image–text pairs have been carefully cleaned. To ensure a fair comparison with SSL-based methods, we pretrain JARA on the open-source Laion2B (Schuhmann et al., 2022) and GRIT12M (Peng et al., 2023) datasets, which are publicly accessible and researchers can used without further annotations or filtering.

## 4.1 ASSESSING CROSS-MODAL UNDERSTANDING VIA IMAGE-TEXT RETRIEVAL

**Comparison.** We benchmark JARA against recently published CLIP-based methods on zero-shot image–text retrieval over COCO2017 (Lin et al., 2014) and Flickr30K (Young et al., 2014). The compared approaches are grouped into four categories as follows: **(A)** image-level CL baselines; **(B)** data-centric enrichment via re-captioning, curation, or distillation; **(C)** SSL-enhanced CLIP that stitches vision-only self-supervised learning into CLIP; and **(D)** our proposed JARA-CLIP.

Tab. 3 shows a complete list of methods under each category. We report results on two encoder scales, ViT-B/16 and ViT-L/14, in Tab. 3 and Tab. 4, respectively. As many recent methods do not provide ViT-L/14 results, we select representative ones or the latest methods (FineCLIP (Jing

Table 3: Zero-shot image-text retrieval results on ViT-B/16. TIPS* is trained on open-sourced Laion2B. ECCV and NeurIPS follow a same-year acceptance and publication cycle. All results in the table correspond to Recall@1(R@1). Performance with I2T/T2I R@5, R@10 is in Appendix B.

| Model | Data Size | COCO val | | Flickr30k *test* | | Flickr30k *full* | |
|---|---|---|---|---|---|---|---|
| | | I2T | T2I | I2T | T2I | I2T | T2I |
| *(A) Image-level CL baselines* | | | | | | | |
| CLIP (Radford et al., 2021) | WIT400M | 51.7 | 32.7 | 84.0 | 71.6 | 44.1 | 24.7 |
| OpenCLIP (Cherti et al., 2023) | Laion2B | 59.4 | 38.9 | 87.1 | 66.2 | 50.0 | 30.6 |
| SigLIP (Zhai et al., 2023) | CC12M | 55.3 | 40.4 | 82.5 | 66.7 | - | - |
| SigLIP (Zhai et al., 2023) | **WebLI-20B** | 62.6 | 44.9 | - | - | - | - |
| RetainCLIP (Feng et al., 2025) | ShareGPT4V | 59.4 | 41.1 | - | - | 52.9 | 34.5 |
| *(B) Data-centric enrichment (re-captioning / curation / distillation)* | | | | | | | |
| FILIP (Yao et al., 2021) | 300M | 40.2 | 29.5 | 69.0 | 55.8 | - | - |
| LongCLIP (Zhang et al., 2024) | ShareGPT4V | 57.6 | 40.4 | - | - | 46.8 | 34.1 |
| DreamLIP (ECCV) (Zheng et al., 2024) | CC12M | 54.0 | 40.6 | 84.3 | 68.3 | - | - |
| TULIP (Najdenkoska et al., 2024) | ShareGPT4V | 56.8 | 40.7 | - | - | 46.1 | 35.2 |
| FineLIP (Asokan et al., 2025) | CC13M | 58.7 | 40.4 | - | - | 52.8 | 34.1 |
| *(C) SSL-enhanced CLIP (self-distillation / MIM stitched into CLIP)* | | | | | | | |
| SimCLR-CLIP (SLIP) (Mu et al., 2022) | CC12M | 37.6 | 26.8 | 62.5 | 46.6 | - | - |
| EVA02-CLIP (Fang et al., 2023) | Merged-2B | 58.7 | 41.7 | 85.9 | 71.5 | 50.3 | 33.8 |
| SF-CLIP (Sameni et al., 2024) | CC12M | 44.3 | 31.4 | 71.8 | 59.9 | - | - |
| CLIPSelf (Wu et al., 2024) | COCO train | 18.8 | 16.1 | 33.8 | 35.0 | - | - |
| FineCLIP (NeurIPS) (Jing et al., 2024) | 13M | 54.5 | 40.2 | 82.5 | 67.9 | - | - |
| SILC (ECCV) (Naeem et al., 2024) | **WebLI-20B** | 62.5 | 44.9 | - | - | - | 31.9 |
| TIPS* (Maninis et al., 2025) | Laion2B | - | - | 82.6 | 67.6 | - | - |
| *(D) Ours* | | | | | | | |
| **JARA-CLIP** | GRIT12M | 60.8 | 44.0 | **87.5** | 72.4 | **53.8** | 34.4 |
| **JARA-CLIP** | Laion2B | **62.6** | **49.0** | 87.6 | **77.1** | 52.3 | **39.6** |

Table 4: Zero-shot image-text retrieval results on ViT-L/14. FG-CLIP* is trained on Laion2B with region captions but without highly detailed image captions.

| Model | Data Size | COCO val | | Flickr30k *test* | | Flickr30k *full* | |
|---|---|---|---|---|---|---|---|
| | | I2T R@1 | T2I R@1 | I2T R@1 | T2I R@1 | I2T R@1 | T2I R@1 |
| OpenCLIP (Cherti et al., 2023) | Laion2B | 59.4 | 43.4 | 85.9 | 73.3 | 48.3 | 35.9 |
| EVA02-CLIP (Fang et al., 2023) | Merged-2B | 64.7 | 47.3 | 89.7 | 78.0 | 59.4 | 43.2 |
| FineLIP (Asokan et al., 2025) | CC13M | 63.4 | 46.2 | - | - | 52.8 | 34.1 |
| FG-CLIP* (Xie et al., 2025) | Laion2B | 65.2 | 47.1 | 90.3 | 79.3 | - | - |
| **JARA-CLIP** | GRIT12M | 64.5 | 48.0 | **92.2** | 78.0 | **60.3** | 43.7 |
| **JARA-CLIP** | Laion2B | **65.8** | **53.2** | 90.5 | **81.0** | 59.2 | **46.8** |

et al., 2024), FG-CLIP* (Xie et al., 2025), and EVA02-CLIP (Fang et al., 2023)) for comparison under large-scale model settings. To ensure fairness, we use OpenCLIP pretrained on Laion2B as the reference baseline and only consider reported results where the baseline performance is consistent with OpenCLIP, since otherwise differences may stem from data preprocessing rather than methodological improvements.

**Results and Scaling.** JARA-CLIP substantially outperforms prior SSL-enhanced CLIP variants (FineCLIP, SILC, TIPS*) and even data-centric CLIP trained on 12M curated data, while learning robust alignments from both clean (GRIT12M) and noisy web data (Laion2B). On COCO T2I with ViT-B/16, R@1 improves from 38.9 (OpenCLIP) → 49.0, compared with 40.2 (FineCLIP) and 44.9 (SILC); on Flickr30K-full, from 30.6 → 39.6, surpassing 34.1 (TIPS*). I2T gains are smaller (*e.g.*, 59.4 (OpenCLIP) → 62.6 on COCO), but the much stronger T2I improvements highlight that patch-level reconstruction enriches regional semantics and strengthens grounding. When scaling to ViT-L/14, JARA's advantages remain consistent: COCO T2I rises to 53.2, and Flickr30K-full to 46.8, even surpassing FG-CLIP* trained on the same Laion2B data with region captions but without highly fine-grained image captions.

Table 5: OVSeg comparison.

| Model | Backbone | ADE-150 mIoU | ADE-150 mAcc | ADE-847 mIoU | ADE-847 mAcc | Pascal Context-59 mIoU | Pascal Context-59 mAcc |
|---|---|---|---|---|---|---|---|
| OVSeg (Liang et al., 2023) | Swin-B/16 | 24.8 | - | 7.1 | - | 53.3 | - |
| SAN (Xu et al., 2023a) | ViT-B/16 | 27.5 | 45.6 | 10.1 | 21.1 | 53.8 | 73.0 |
| Cat-Seg (Xu et al., 2023b) | ViT-B/16 | 27.2 | 41.2 | 8.4 | 16.6 | 57.5 | 74.0 |
| DeOP (Han et al., 2023) | ViT-B/16 | 22.9 | - | 7.1 | - | 48.8 | - |
| EBSeg (Shan et al., 2024) | ViT-B/16 | 30.0 | - | 11.7 | - | 56.7 | - |
| SED (Xie et al., 2024) | ConvNeXt-B | 31.6 | - | 11.4 | - | 57.3 | - |
| CLIPSelf (Wu et al., 2024) | ViT-B/16 | 29.7 | 45.1 | 10.1 | 17.2 | 55.3 | 73.4 |
| FineCLIP (Jing et al., 2024) | ViT-B/16 | 32.4 | **50.5** | 12.2 | 22.2 | 56.0 | 74.4 |
| HyperCLIP (Peng et al., 2025) | ViT-B/16 | 30.5 | - | 10.3 | - | 57.6 | - |
| JARA-CLIP | ViT-B/16 | **33.0** | 49.8 | **12.8** | **23.0** | **58.3** | **76.5** |

Table 6: Comparisons on computational overhead.

| Model | Contrast | Recon. | Fwd FLOPs | Fwd-Bwd FLOPs | Lantency / step | Memory |
|---|---|---|---|---|---|---|
| CLIP | ✓ | × | 33.7G | 101.1G | 784ms | 32GB |
| EVA-CLIP (CL only) | ✓ | × | 33.8G | 101.3G | 805ms | 32GB |
| EVA-CLIP (MIM only) | × | ✓ | 581.7G | 1219.8G | 2418ms | 56GB |
| JARA-CLIP | ✓ | ✓ | 57.7G | 105.8G | 1105ms | 52GB |

## 4.2 ASSESSING DENSE REPRESENTATIONS VIA SEGMENTATION

**Open-Vocabulary Segmentation (OVSeg).** We follow the standard OVSeg setup by fine-tuning on COCO-Stuff base classes and evaluating on novel categories of ADE20K (Zhou et al., 2019) and PASCAL Context (Mottaghi et al., 2014), using Cat-Seg (Xu et al., 2023b) to adapt CLIP for segmentation. As shown in Tab. 5, JARA-CLIP consistently achieves the best performance. Compared to other SSL-enhanced CLIP variants, JARA yields stronger dense representations: On ADE20K, it improves mIoU from 12.2 (FineCLIP) → 12.8.

Notably, JARA surpasses CLIPSelf—which relies on CLS-token distillation—by a clear margin (58.3 vs. 55.3 mIoU), showing that SSL can substantially improve the quality of dense representations rather than merely injecting extra knowledge. These results show that patch-level reconstruction substantially enhances dense representations, outperforming fine-grained contrastive learning (FineCLIP), and distillation (CLIPSelf). Thus, JARA's SSL can genuinely improve CLIP's dense alignment quality, not just provide auxiliary knowledge.

**Zero-Shot Segmentation (ZSSeg).** We assess the intrinsic concept that model learned during VLP with ZSSeg task. Following the MaskCLIP (Zhou et al., 2022) protocol, we evaluate JARA's frozen backbone directly on the COCO validation set. As shown in Tab. 2b, JARA improves mIoU from 12.5% to 13.1% and aAcc from 23.7% to 26.0% on ViT-B. With ViT-L, the gains are larger, with mIoU up from 8.3% to 10.6% and aAcc from 11.1% to 19.1%. These results confirm JARA's stronger transfer ability in segmenting unseen categories.

Table 7: Comparison of different JARA pretraining designs. Per-step Latency is measured as the total per-batch compute time (forward + backward) with a batch size of 28.6k.

| Method | Input Image Type Context Img | Input Image Type Full Img | Image Representation for CL [CLS] Token | Image Representation for CL Dense Token Pooling | Shared Student Encoder CL & REC |
|---|---|---|---|---|---|
| Context-Pool (ROIAlign) | ✓ | × | × | ✓ (ROIAlign) | ✓ |
| Context-Pool (Attention) | ✓ | × | × | ✓ (AttnPool) | ✓ |
| Full-CLS-Shared (CL+REC) | × | ✓ | ✓ | × | ✓ |
| Full-CLS-Dual (CL/REC) | × | ✓ | ✓ | × | × |
| Context-CLS | ✓ | × | ✓ | × | ✓ |

| Method | ImageNet1K Acc@1 | COCO I2T R@1 | COCO T2I R@1 | Flickr30K I2T R@1 | Flickr30K T2I R@1 | Latency ms/step |
|---|---|---|---|---|---|---|
| Context-Pool (ROIAlign) | 63.5 | 50.8 | 33.8 | 80.6 | 61.8 | **2185** |
| Context-Pool (Attention) | 14.6 | - | - | - | - | 4070 |
| Full-CLS-Shared (CL+REC) | 68.7 | 60.8 | 45.3 | 86.8 | 72.9 | 4100 |
| Full-CLS-Dual (CL/REC) | 68.3 | 62.4 | **49.1** | 87.2 | 75.2 | 3910 |
| Context-CLS | **69.0** | **62.6** | 49.0 | **87.6** | **77.1** | 2400 |

### 4.3 ABLATION STUDY

We systematically dissect the JARA framework to analyze the contribution of its key components and design choices. All ablations are conducted using the ViT-B model.

**Ablation on the JARA-style MIM and CMSD. (a) Necessity:** Tab. 8's first row shows removing JARA-Style MIM entirely results in a marked performance degradation and we gain the best perfomance when $\lambda = 2$. Tab. 2a shows that compared with simply conduct CLIP's full image-text contrastive learning, CMSD has better performance (Flickr30k-test T2I R@1: $66.2 \rightarrow 68.2$). And when JARA-Style MIM is sequentially added to CMSD, performance gains (Flickr30k-test R@1 74.5: $68.2 \rightarrow 68.2$). **(b) Loss Function:** Tab. 1b and Fig. 3c compare different reconsturction functions. Smooth L1 strikes the best balance between final performance and training stability.

**Ablation on JARA's pretraining designs.** We analyze the optimal strategy to derive image embedding for Contrastive Learning. As shown in Tab. 7, the representation by pooling dense tokens from the visible context (`Context-Pool`) is suboptimal: both **(1) ROIAlign** (He et al., 2017) and **(2) Attention Pool** (Naeem et al., 2024; Yu et al., 2022) perform poorly. Using the `[CLS]` token from the full image introduces a different challenge. A shared embedding for CL and local reconstruction **(3) Full-CLS-Shared** creates an optimization conflict between different objectives, which degrades performance and incurs high train-step latency (4100 ms). A dual-pass approach **(4) Full-CLS-Dual** mitigates this conflict, slightly improving I2T retrieval, but at the cost of a prohibitive latency (3910 ms). Our proposed **(5) Context-CLS** strategy, using the `[CLS]` token from the visible context, resolves these issues. It achieves a superior 69.0 ImageNet1k zero-shot classification Acc@1 and 49.0 T2I R@1 on COCO with a low train-step latency of 2400 ms, establishing it as the most effective and efficient approach.

Table 8: Ablation on $\lambda$, *i.e.*, REC:CL.

| Model | Dataset | REC:CL | ImageNet Acc@1 | COCO I2T R@1 / T2I R@1 | Flickr30K I2T R@1 / T2I R@1 |
|-------|---------|--------|---------|---------|---------|
| JARA-CLIP | Laion | 0:1 (w/o REC) | 67.9 | 60.1 / 41.4 | 87.2 / 68.2 |
| JARA-CLIP | Laion | 1:2 | 68.6 | 62.0 / 48.4 | 87.3 / 75.0 |
| JARA-CLIP | Laion | 1:1 | 68.5 | 61.6 / 48.9 | 87.1 / 76.6 |
| JARA-CLIP | Laion | 2:1 | **69.0** | **62.6 / 49.0** | **87.6 / 77.1** |

**Ablation on SBM.** We ablate SBM on OVSeg and ZSSeg. As shown in Tab. 1a, **(1) Strict Uniform Masking** (Li et al., 2022) hinders local context learning due to rigid sampling, while **(2) DINO-style masking** suffers from central bias (Fig. 3a). In contrast, our **(3) SBM** achieves uniform spatial coverage and preserves context (Fig. 3b), yielding the best performance with coherent dense representations. We acknowledge this to SBM improves central and peripheral regions' coherence, and we also visualize this effect in Fig. 4.

**Training-time computational overhead.** As shwon in Tab. 6, EVA-CLIP adds MIM phase upon CLIP and we can find a heavily increase on both Fwd flops ($\times$ 16.3) and Fwd-Bwd flops ($\times$ 11.1). But JARA-CLIP, where we only add ($\times$ 0.7) Fwd flops and $\times$ 0.04 Fwd-Bwd Flops. Further details and analysis are provided in Appendix E.

## 5 CONCLUSION

We presented JARA, a unified SSL-enhanced CLIP framework for region-aware VLP. Trained solely on Laion2B and GRIT12M, it achieves strong results on OVSeg and retrieval, and also shows consistent improvements on ZSSeg. These results highlight its robustness to noisy web data and effectiveness for fine-grained dense representation learning. As future work, we plan to extend JARA to region-concept learning and detection for stronger structured vision-language understanding.

ETHICS STATEMENT

This work complies with the ICLR Code of Ethics and Code of Conduct. It does not involve experiments with humans, crowdsourced workers, animals, or the use of sensitive personal information (such as identifiable faces, medical records, or financial data), and therefore does not require IRB approval. All datasets employed are publicly released for research and were used strictly within the terms of their respective licenses. No restricted data were redistributed.

REPRODUCIBILITY STATEMENT

We provide detailed descriptions of the model architecture, loss functions, and hyperparameter settings to facilitate reproducibility. After the review process, we will release resources in two phases: (1) model checkpoints and test scripts; (2) full source code and train scripts.

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

## A    JARA'S PRETRAINING DETAILS

Table 9: Training configurations for ViT-B/16 and ViT-L/14@336.

| | Setting | ViT-B/16 | ViT-L/14@336 |
|---|---|---|---|
| 0 | Backbone | ViT-B/16 | ViT-L/14 |
| 1 | Input Resolution | 224×224 | 336×336 |
| 2 | Batch Size Per NPU | 448 | 80 |
| 3 | Seen Samples | 2B | 2B |
| 4 | Predictor Depth | 6 | 6 |
| 5 | Predictor Embedding Dim | 384 | 384 |
| 6 | Learning Rate | 1e-5 | 1e-5 |
| 7 | Weight Decay | 0.01 | 0.01 |
| 8 | Warmup Ratio | 0.03 | 0.03 |
| 9 | EMA | 0.996–1.0 | 0.996–1.0 |
| 10 | Final LR | 5e-9 | 5e-9 |
| 11 | Final Weight Decay | 0.05 | 0.1 |
| 12 | Gradient Checkpointing | ✓ | ✓ |
| 13 | bfloat16 | ✓ | ✓ |
| 14 | Patch Size | 16 | 14 |
| 15 | World Size (NPUs) | 64 | 160 |
| 16 | Total Batch Size | 28.6k | 12.8k |

## B    ZERO-SHOT RETRIEVAL COMPARISON

Here we provide the full comparison data including R@1,R@5 and R@10.

Table 10: Zero-shot image-text retrieval results on COCO. "Data Size" column indicates the scale of pretraining data; JARA is trained solely on LAION-2B and GRIT-12M.

| | Model | Data Size | I2T R@1 | R@5 | R@10 | T2I R@1 | R@5 | R@10 |
|---|---|---|---|---|---|---|---|---|
| ViT-B/16 | CLIP (Radford et al., 2021) | 400M | 51.7 | 76.7 | 84.3 | 32.7 | 57.8 | 68.3 |
| | LongCLIP (Zhang et al., 2024) | ShareGPT4v | 57.6 | 81.1 | 87.8 | 40.4 | 65.8 | 75.2 |
| | OpenCLIP (Cherti et al., 2023) | Laion(2B) | 59.4 | 81.6 | 88.3 | 38.9 | 64.1 | 73.6 |
| | EVA02-CLIP (Fang et al., 2023) | Merged-2B | 58.7 | 81.0 | 88.5 | 41.7 | 66.3 | 75.7 |
| | FILIP (Yao et al., 2021) | 300M | 40.2 | 66.0 | 76.3 | 29.5 | 55.3 | 66.3 |
| | CLIPSelf (Wu et al., 2024) | COCO train | 18.8 | 38.9 | 50.4 | 16.1 | 34.5 | 45.1 |
| | FineCLIP (Jing et al., 2024) | 13M | 54.5 | 78.6 | 85.8 | 40.2 | 66.5 | 76.1 |
| | TULIP (Najdenkoska et al., 2024) | ShareGPT4v | 56.8 | 80.3 | - | 40.7 | 66.1 | - |
| | FineLIP (Asokan et al., 2025) | CC13M | 58.7 | 81.4 | 88.2 | 40.4 | 66.2 | 76.0 |
| | **JARA-CLIP** | GRIT(12M) | 60.8 | 83.5 | 89.4 | 44.0 | 69.5 | 78.9 |
| | **JARA-CLIP** | Laion(2B) | **62.6** | **84.4** | **90.6** | **49.0** | **74.3** | **82.9** |
| ViT-L/14 | CLIP (Radford et al., 2021) | 400M | 58.5 | 80.7 | 87.7 | 36.1 | 60.5 | 70.7 |
| | LongCLIP (Zhang et al., 2024) | ShareGPT4v | 62.8 | 85.1 | 91.2 | 46.3 | 70.8 | 79.8 |
| | OpenCLIP (Cherti et al., 2023) | Laion(2B) | 59.4 | 82.4 | 89.0 | 43.4 | 68.5 | 77.6 |
| | FineLIP (Asokan et al., 2025) | CC13M | 63.4 | 84.8 | 91.0 | 46.2 | 71.2 | 80.1 |
| | TULIP (Najdenkoska et al., 2024) | ShareGPT4v | 62.6 | 84.7 | - | 46.1 | 71.1 | - |
| | EVA02-CLIP (Fang et al., 2023) | Merged-2B | 64.7 | 85.3 | 90.8 | 47.3 | 71.0 | 79.7 |
| | **JARA-CLIP** | GRIT(12M) | 64.5 | 85.7 | 91.1 | 48.0 | 72.3 | 81.1 |
| | **JARA-CLIP** | Laion(2B) | **65.3** | **86.7** | **92.0** | **52.7** | **77.5** | **85.0** |

Table 11: Zero-shot image-text retrieval results on Flickr30k. "Data Size" column indicates the scale of pretraining data; JARA is trained solely on LAION-2B and GRIT-12M.

| | Model | Data Size | Flickr30k *test* | | | | | | Flickr30k *full* | | | | | |
|---|---|---|---|---|---|---|---|---|---|---|---|---|---|---|
| | | | I2T | | | T2I | | | I2T | | | T2I | | |
| | | | R@1 | R@5 | R@10 | R@1 | R@5 | R@10 | R@1 | R@5 | R@10 | R@1 | R@5 | R@10 |
| ViT-B/16 | CLIP (Radford et al., 2021) | 400M | 84.0 | 96.1 | 98.2 | 71.6 | 90.3 | 94.1 | 44.1 | 68.2 | 77.0 | 24.7 | 45.1 | 54.6 |
| | LongCLIP (Zhang et al., 2024) | ShareGPT4v | - | - | - | - | - | - | 46.8 | 71.4 | - | 34.1 | 56.3 | - |
| | OpenCLIP (Cherti et al., 2023) | Laion(2B) | 87.1 | **98.1** | 99.2 | 66.2 | 87.8 | 93.1 | 50.3 | 74.1 | 81.9 | 33.8 | 56.3 | 65.3 |
| | EVA02-CLIP (Fang et al., 2023) | Merged-2B | 85.9 | 96.6 | 98.8 | 71.5 | 91.1 | 94.7 | 50.3 | 74.1 | 81.9 | 33.8 | 56.3 | 65.3 |
| | FILIP (Yao et al., 2021) | 300M | 69.0 | 89.8 | 94.0 | 55.8 | 81.5 | 96.3 | - | - | - | - | - | - |
| | CLIPSelf (Wu et al., 2024) | COCO train | 33.8 | 61.7 | 73.0 | 35.0 | 61.3 | 32.7 | - | - | - | - | - | - |
| | FineCLIP (Jing et al., 2024) | 13M | 82.5 | 96.4 | 98.6 | 67.9 | 89.1 | 94.1 | - | - | - | - | - | - |
| | TULIP (Najdenkoska et al., 2024) | ShareGPT4v | - | - | - | - | - | - | 46.1 | 70.8 | - | 35.2 | 57.2 | - |
| | FineLIP (Asokan et al., 2025) | CC13M | - | - | - | - | - | - | 52.8 | 75.3 | 82.9 | 34.1 | 56.7 | 65.8 |
| | **JARA-CLIP** | GRIT(12M) | **87.5** | 97.4 | **99.5** | 72.4 | 90.9 | 94.8 | **53.8** | **76.6** | **84.1** | 34.4 | 56.6 | 65.6 |
| | **JARA-CLIP** | Laion(2B) | 87.6 | 97.8 | 99.2 | **77.1** | **93.9** | **97.0** | 52.3 | 76.2 | 84.0 | **39.6** | **63.1** | **71.9** |
| ViT-L/14 | CLIP (Radford et al., 2021) | 400M | 87.7 | 98.5 | 99.4 | 66.9 | 89.0 | 93.4 | 48.5 | 72.6 | 80.8 | 28.0 | 49.3 | 58.7 |
| | LongCLIP (Zhang et al., 2024) | ShareGPT4v | - | - | - | - | - | - | 53.4 | 77.5 | - | 41.2 | 64.1 | - |
| | OpenCLIP (Cherti et al., 2023) | Laion(2B) | 85.9 | 97.9 | 99.1 | 73.3 | 91.7 | 95.1 | 48.3 | 73.6 | 82.1 | 35.9 | 58.4 | 67.3 |
| | FineLIP (Asokan et al., 2025) | CC13M | - | - | - | - | - | - | 52.8 | 75.3 | 82.9 | 34.1 | 56.7 | 65.8 |
| | TULIP (Najdenkoska et al., 2024) | ShareGPT4v | - | - | - | - | - | - | 56.7 | 79.5 | - | 35.2 | 57.4 | - |
| | EVA02-CLIP (Fang et al., 2023) | Merged-2B | 89.7 | **98.9** | 99.6 | 78.0 | 94.3 | 96.8 | 59.4 | 81.9 | 88.5 | 43.2 | 65.7 | 73.8 |
| | **JARA-CLIP** | GRIT(12M) | **92.2** | 98.8 | 99.4 | 78.0 | 94.9 | 97.3 | **60.3** | **82.5** | **88.9** | 43.7 | 66.3 | 74.5 |
| | **JARA-CLIP** | Laion(2B) | 90.5 | 98.6 | **99.6** | **81.0** | **95.7** | **97.8** | 59.2 | 81.7 | 88.3 | **46.8** | **69.8** | **78.0** |

---

**Algorithm 1** Spatially Balanced Masking (SBM)

---

**Input:** Image patch grid $\{p_i\}_{i=1}^{H \times W}$, historical mask counts $\{F_i\}$, region size $(h, w)$, neighborhood size $k$

**Output:** Masked patch indices $\mathcal{M}$

/* Patch Frequency–Aware Sampling */

1: **for** each patch $i$ **do**
2:     $s_i \leftarrow \frac{1}{1+\exp(F_i)}$         ▷ sigmoid transform of historical count
3: **end for**
4: $w_i \leftarrow s_i / \sum_{j=1}^{H \times W} s_j$     ▷ normalize into probability distribution
5: Sample candidate patch $i$ according to $w_i$
6: Convert index $i$ into 2D grid location $(x, y)$
    /* Offset-Aware Region Sampling */
7: Compute neighborhood average frequency $\bar{F}_n$ within $k \times k$ around $(x, y)$
8: $\delta_x \sim \text{Uniform}\left(-\frac{h}{2}(1 + \bar{F}_n), \frac{h}{2}(1 + \bar{F}_n)\right)$
9: $\delta_y \sim \text{Uniform}\left(-\frac{w}{2}(1 + \bar{F}_n), \frac{w}{2}(1 + \bar{F}_n)\right)$
10: $x_c \leftarrow x + \delta_x, \quad y_c \leftarrow y + \delta_y$
    /* Define final mask region */
11: $\text{Mask}_{pos} \leftarrow [x_c, y_c, h, w]$
12: Convert rectangular mask $\text{Mask}_{pos}$ into patch index set $\mathcal{M}$
13: **return** $\mathcal{M}$

---

## C DETAILS AND VISUALIZATION FOR SBM

Existing masked modeling strategies often adopt random sampling or fixed-size box cropping, which tends to over-sample the central region of an image. This leads to spatial imbalance, where central regions are repeatedly masked while peripheral patches remain underutilized, hindering uniform spatial representation learning. To address this issue, we introduce Spatially Balanced Masking (SBM) — a dynamic sampling mechanism that ensures a more uniform spatial distribution of masked patches throughout training.

**Patch Frequency–Aware Sampling.** Let $F_i$ denote the historical masking count of patch $i$, tracked across training steps. We compute its sampling score using a sigmoid-shaped transformation:

$$s_i = \frac{1}{1 + \exp(F_i)}. \tag{7}$$

We normalize all scores across the full $H \times W$ grid to form a valid sampling distribution:

$$w_i = \frac{s_i}{\sum_{j=1}^{H \times W} s_j}. \tag{8}$$

This design increases the likelihood of selecting patches that have rarely been masked, leading to more spatially balanced supervision across epochs.

**Offset-Aware Region Sampling.** To prevent frequent masking in local clusters, we incorporate a neighborhood-aware offset mechanism when selecting a target masking region. Specifically, given a target crop size $(h, w)$, we first sample a candidate patch $i$ using weights $w_i$, and convert it into a 2D grid location $(x, y)$. We then compute the average masking frequency $\bar{F}_n$ within a square neighborhood of size $k \times k$ centered at $(x, y)$. The sampling offset range is dynamically adjusted as:

$$\delta_x \sim \text{Uniform}\left(-\frac{h}{2} \cdot (1 + \bar{F}_n), \frac{h}{2} \cdot (1 + \bar{F}_n)\right), \tag{9}$$

$$\delta_y \sim \text{Uniform}\left(-\frac{w}{2} \cdot (1 + \bar{F}_n), \frac{w}{2} \cdot (1 + \bar{F}_n)\right), \tag{10}$$

where $\delta_x$, $\delta_y$ are applied to $(x, y)$ to shift the center of the masked region. Finally, the center of the masked region is adjusted by the sampled offsets, and the resulting rectangular mask is defined as:

$$\text{Mask}_{\text{pos}} = [x_c, y_c, h, w], \tag{11}$$

where $x_c = x + \delta_x$, $y_c = y + \delta_y$ denote the center coordinates after offset, and $(h, w)$ is the region size. The final mask is then converted into a list of patch indices within this rectangular area.

This dynamic offset mechanism ensures that frequently masked regions are pushed outward, while underrepresented areas are sampled with higher probability and lower perturbation. As a result, SBM provides a more balanced and adaptive coverage of the image space, facilitating better region-aware visual learning. The pseudocode is displayed in Algorithm 1.

**Visualization of the effect of SBM.** Fig. 4 compares JARA-CLIP with and without SBM. With SBM, PCA visualizations exhibit more coherent and semantically consistent features across the entire image, particularly in peripheral regions such as corners. The attention maps also become cleaner and more focused, indicating that SBM reduces background noise and sharpens attention on salient regions.

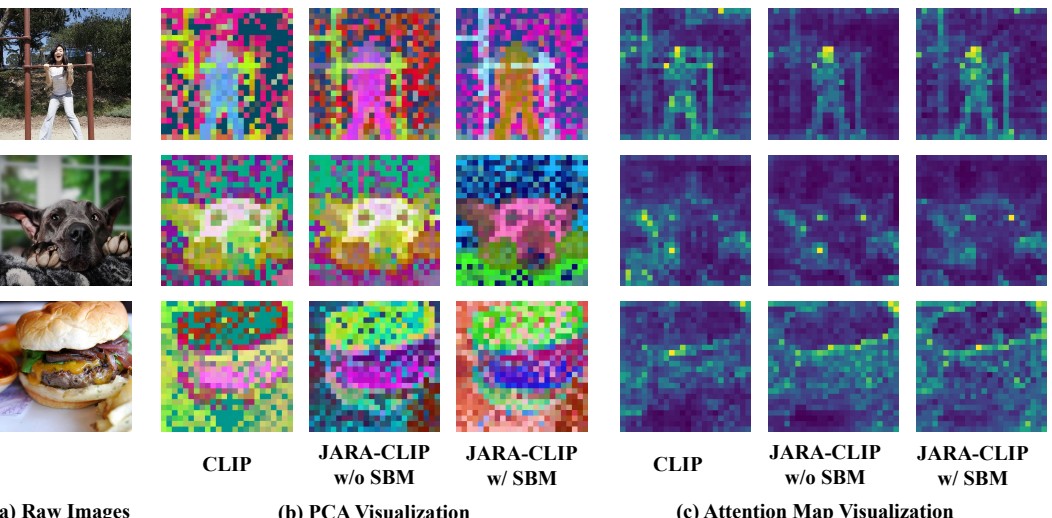

| | CLIP | JARA-CLIP w/o SBM | JARA-CLIP w/ SBM | CLIP | JARA-CLIP w/o SBM | JARA-CLIP w/ SBM |

(a) Raw Images    (b) PCA Visualization    (c) Attention Map Visualization

Figure 4: Comparison for the PCA visualization and attention map visualization for JARA-CLIP w/ SBM and w/o SBM.

## D  MORE COMPARISON FOR RESULTS ON VIT-L/14

Tab. 12 presents additional results on ViT-L/14, including comparisons with a broader set of state-of-the-art methods. JARA-CLIP continues to demonstrate strong retrieval performance, particularly on T2I tasks, highlighting the scalability potential of the JARA framework.

Table 12: Zero-shot image-text retrieval results on ViT-L/14. FG-CLIP* and TIPS* does not use additional data. JARA is trained solely on LAION-2B and GRIT-12M **without** region locating data or additional pipeline for data cleaning. ECCV and NeurIPS's accepted year is same as their published year. All results in the table correspond to Recall@1(R@1).

| Model | Data Size | COCO val | | Flickr30k *test* | | Flickr30k *full* | |
|---|---|---|---|---|---|---|---|
| | | I2T | T2I | I2T | T2I | I2T | T2I |
| *(A) Image-level CL baselines* | | | | | | | |
| CLIP (Radford et al., 2021) | WIT400M | 58.5 | 36.1 | 87.7 | 66.9 | 48.5 | 28.0 |
| OpenCLIP (Cherti et al., 2023) | Laion2B | 59.4 | 43.4 | 85.9 | 73.3 | 48.3 | 35.9 |
| RetainCLIP (Feng et al., 2025) | ShareGPT4V | 64.4 | 47.0 | - | - | 60.2 | 42.2 |
| *(B) Data-centric enrichment* | | | | | | | |
| LongCLIP (Zhang et al., 2024) | ShareGPT4V | 62.8 | 46.3 | - | - | 53.4 | 41.2 |
| TULIP (Najdenkoska et al., 2024) | ShareGPT4V | 62.6 | 46.1 | - | - | 56.7 | 35.2 |
| FineLIP (Asokan et al., 2025) | CC13M | 63.4 | 46.2 | - | - | 52.8 | 34.1 |
| FG-CLIP* (Xie et al., 2025) | Laion2B | 65.2 | 47.1 | 90.3 | 79.3 | - | - |
| *(C) SSL-enhanced CLIP (self-distillation / MIM stitched)* | | | | | | | |
| EVA02-CLIP (Fang et al., 2023) | Merged-2B | 64.7 | 47.3 | 89.7 | 78.0 | 59.4 | 43.2 |
| *(D) Ours* | | | | | | | |
| **JARA-CLIP** | GRIT12M | 64.5 | 48.0 | **92.2** | 78.0 | **60.3** | 43.7 |
| **JARA-CLIP** | Laion2B | **65.8** | **53.2** | 90.5 | **81.0** | 59.2 | **46.8** |

## E  ANALYSIS OF TRAINING-TIME COMPUTATIONAL OVERHEAD

We compared JARA,CLIP,and EVA-CLIP using ViT-B. For a fair comparison, JARA's end-to-end training is benchmarked against EVA-CLIP's full two-stage pipeline (MIM pretraining + contrastive learning), and all models are trained without grad-checkpointing. We calculated the vision tower FLOPs using 'fvcore' and cross-verified the results with 'calflops'. We calculated train time and memory by training on 16 Ascend 910B GPUs, with a total batch size of 3584.

As shown in Tab. 6, JARA's overhead is only marginally higher than CLIP. This efficiency is achieved by: **(1)Partial Input.** The student encoder processes only 50% patches (context), significantly reducing its computational load, resulting in an average of 16.8G Forward FLOPs. SBM is crucial for this alignment's stability. **(2) Lightweight Components.** The predictor is a small 6-layer ViT, contributing only 7.1G Forward FLOPs. And the teacher encoder doesn't require grad during bp(updated via EMA), which minimizes extra costs.Meanwhile EVA-02 shows a significant increase in these metrics because its use EVA_CLIP_g (1B) as teacher.

## USE OF LARGE LANGUAGE MODELS

A large language model (LLM) was used solely for language-level assistance, such as improving readability, fluency of the text and formatting LaTeX tables and retrieve related works. The research ideas, experiments, and results are entirely the work of the authors, who bear full responsibility for the content of this submission.

