# OpenReview forum: "JARA: Joint Alignment and Reconstruction Architecture for Region-Aware Vision-Language Pretraining"
_ICLR.cc/2026/Conference — Submitted to ICLR 2026_

### Official Review · Reviewer_vB4j · 2025-10-31

**Soundness:** 2
**Presentation:** 1
**Contribution:** 2
**Rating:** 2
**Confidence:** 4

**Summary:**

This paper proposes JARA, a framework that enables region-aware vision-language pretraining via a spatially balanced masking strategy. The approach aims to improve CLIP’s region-level understanding via a self-supervised learning setup.

**Strengths:**

- The paper demonstrates strong performance on large-scale training, showing improvements in zero-shot retrieval and zero-shot segmentation tasks.

**Weaknesses:**

- **Lack of systematic comparison.**
Across multiple tables, the experimental configurations vary widely in terms of backbone architecture and dataset scale, making it difficult to assess whether the proposed framework truly outperforms prior work. While reproducing every baseline under identical conditions may be infeasible, major recent methods trained on public datasets should be compared under consistent settings—especially given the strong influence of batch size on contrastive learning performance.

- **Incomplete comparison for zero-shot segmentation.**
In Table 2(b), the model is compared only against CLIP, which is expected to underperform by design. Other recent segmentation approaches are missing. Furthermore, Table 5 compares with OVSeg and others that seem to use different datasets, and some tables inconsistently report data size or backbone only. Even if standardized, such implementation details should be explicitly described in the table captions for transparency.

- **Poor paper organization.**
The structure is difficult to follow, and the correspondence between text and tables is inconsistent. For example, Section 4.2 discusses Tables 5 and 6 before suddenly referencing Table 2(b). This makes the experimental analysis hard to trace.

- **Presentation of results is overly numerical.**
The results section reads more like a report than a research paper, listing performance numbers without sufficient discussion or insight. The method’s novelty also appears limited from a technical standpoint.

- **Complex and unclear figures.**
The figures are overly detailed, making it difficult to grasp the core idea of the method at a glance. The illustrations should more clearly convey how the proposed architecture improves region-aware learning compared to existing approaches.

**Questions:**

- Have the authors evaluated zero-shot classification beyond ImageNet? It is standard in this field to benchmark across multiple datasets—what is the reason for omitting such comparisons?

- In Line 442, the notation “R@1 74.5: 68.2 → 68.2” is unclear. What does this represent?

- To ensure fair comparison, should the experimental tables not include information such as data size, backbone, and batch size for each method? Even if epochs differ, such details are critical to judge the fairness of the evaluation.

---

### Official Review · Reviewer_Xuju · 2025-10-31

**Soundness:** 2
**Presentation:** 2
**Contribution:** 1
**Rating:** 2
**Confidence:** 2

**Summary:**

This paper introduces a novel Vision-Language Pretraining approach that integrates the training methodologies of DINO and CLIP into a unified framework, thereby enhancing the model's performance in downstream applications.

**Strengths:**

1. This paper proposes a masking strategy that mitigates the adverse effects caused by random masking.
2. The paper surpasses existing models with equivalent training volumes in terms of performance on downstream tasks.

**Weaknesses:**

1. The primary issue with this paper is its lack of novelty. The core methodology involves combining DINO's self-distillation approach with CLIP's contrastive learning technique. However, numerous papers referenced in Table 4 have previously explored similar ideas.
2. The mask optimization strategy proposed in the paper is also quite straightforward.
3. The comparison in Table 4 of the paper is not entirely fair, as it lacks a balanced comparison with similar methods under equivalent settings.
4. The images in Figure 2 are somewhat confusing, as some patches that are supposed to be masked still appear in the input for the student model.

**Questions:**

Refer to weaknesses.

---

### Official Review · Reviewer_BtuT · 2025-11-02

**Soundness:** 3
**Presentation:** 2
**Contribution:** 2
**Rating:** 2
**Confidence:** 4

**Summary:**

The paper proposes JARA, a unified framework that enhances CLIP by integrating region-aware learning through a self-supervised objective. JARA uses Spatially Balanced Masking (SBM) for Cross-Modal Self-Distillation through reconstruction of masked patches, replacing self-distillation. JARA achieves competitive results in fine-grained retrieval and segmentation.

**Strengths:**

- JARA combines contrastive learning with masked language modeling supervised by a teacher model. It is an interesting direction to combine vision-language learning with self-supervision.
- SBM is a neat improvement over existing patch cropping strategies that noticeably increases performance.
- The model performs well on multiple tasks including retrieval and semantic segmentation.

**Weaknesses:**

- The presentation of the paper is lacking.
  * In 3.2, the terminology for the losses is not coherent. The loss is defined as $L = L_{\text{CL}} + \lambda L_{\text{REC}}$. But $L_{\text{CL}}$ and $L_{\text{REC}}$ are never defined. Instead, new terms $L_{\text{I2T}}$ and $L_{\text{T2I}}$ are introduced. Every term should be clearly defined in the main text.
  * In 4.3, it is not explained what the ablations Full-CLS-Shared and Full-CLS-Dual refer to.
  * Tables are scatted throughout the paper and often do not appear when they are being referred to. For instance, Tab. 2 appears on page 6 and is referred to on page 8 after Tab. 3, 4, and 5.

- Some methodological choices are not well justified or ablated.
  * As evident from the ablation in Tab. 8, when the reconstruction loss is dropped, the performance does not significantly degrade. This is surprising as this is the core technical contribution of the paper. Without it, the training defaults to the CLIP loss. This seems to significantly weaken this contribution and questions whether the extra compute for the teacher is necessary.
  * It is not clear why the text encoder is frozen. This seems to be a limitation to scale beyond the initial CLIP encoder. An ablation would have helped better understand this choice.
  * In lines 226-228, it is described that the loss is re-weighted for partial labels. However, there is no details on how this is implemented. Specifically, the introduction states that the proposed method does not rely on curated or annotated region-level data.

- The experimental evaluation is incomplete.
  * The paper claims state-of-the-art, but fails to compare to COSMOS [A] which performs better on retrieval and zero-shot semantic segmentation. Due to the high relevance to the paper, COSMOS should be included in the discussion of related works as well.
  * The paper does not compare zero-shot classification performance between the baselines and JARA. ImageNet performance in provided but not put into perspective. New models are typically also evaluated on additional fine-grained classification datasets, such as CIFAR-100, CUB, StanfordCars, etc.
  * Despite the focus on improving fine-grained tasks, there is little experimental evaluation in this regard. Except for semantic segmentation, the other tasks are coarse-grained, and the zero-shot semantic segmentation only compares to CLIP. Qualitative examples could further illustrate differences in the learned representation.

[A] Kim et. al., COSMOS: Cross-Modality Self-Distillation for Vision Language Pre-training, CVPR 2025

**Questions:**

Please refer to the weaknesses section. Questions of particular interest are:
- What are Full-CLS-Shared and Full-CLS-Dual?
- Which of the technical changes from CLIP to JARA contribute the most to the improvements? What is the significance of the REC loss?
- Why is the text encoder frozen?
- How is the loss re-weighting for partial labels implemented? Are the image-text pairs labeled whether they are noisy (partial) or complete?
- How does JARA compare to COSMOS, both in terms of methodology (contributions) and performance-wise?
- How does JARA compare to the other baselines in zero-shot classifiction, at the very least on ImageNet?
- What does the asterisk (*) mean in 4.1 and some table captions?

I am willing to increase my score if my concerns are well addressed.

---

### Official Review · Reviewer_zXfY · 2025-11-06

**Soundness:** 1
**Presentation:** 2
**Contribution:** 1
**Rating:** 2
**Confidence:** 5

**Summary:**

This paper introduces JARA (Joint Alignment and Reconstruction Architecture), a region-aware learning application to CLIP via self-supervised objectives. JARA is a combination of Teacher-Student distillation where the non-gradient EMA updated teacher is fed with the original image, the student is fed with the SBM masked image, the contrastive loss is given between the student [CLS] and text encoder. It follows the ordinary pipeline that joins SSL, T-S distillation with CLIP.

**Strengths:**

The paper is well organized and easy on the eyes.

While there are multiple works in joining self-supervised learning with CLIP, the main contribution of this paper is to develop a cross-modal alignment framework to alleviate the limitations in text agnostic contrastive learning.

I strongly support the idea that CLIP needs multimodal alignment.
I think a step towards this direction will be a breakthrough inbetween the heavy multimodal cross-modality alignments in concatenation-based VLMs and text agnostic yet independent and lightweighted CLIP alignment.

**Weaknesses:**

My biggest concern is the novelty of this paper. Not only that SSL+CLIP [1], distillation+CLIP [2,3,4,5], and multimodal alignment for contrastive loss [6] has been previously proposed, but the term of masking semantic components of the image and distilling it with a teacher-student network which has been proposed in this paper seems to have no novel components.

My second concern is that it does not actually provide Cross-modal alignments.
The base drawbacks in CLIP-wise method is that it is not actually an element-to-element alignment of vision/language tokens, but it is rather the alignment for the abstraction (the [CLS] or [EOS] token in vision/text encoders).
While this paper stresses that it integrates the multi-modality into a single forward path, the alignment is still done in a text agnostic manner, keeping the limitations of previous text agnostic embedding.
In my personal opinion, I think this paper in its current version lacks improvement in the huge complexity computation generated from considering all the image-text token pairs for contrastive losses [7] or VLMs.

[1] SLIP: Self-supervision meets Language-Image Pretraining

[2] MaskCLIP: Masked Self-Distillation Advances Contrastive Language-Image Pretraining, CVPR'23

[3] Expediting Contrastive Language-Image Pretraining via Self-distilled Encoders, AAAI'24

[4] Misalign, Contrast, then Distill: Rethinking Misalignments in Language-Image Pretrianing, ICCV'23

[5] COSMOS: Cross-Modality Self-distillation for Vision Language Pre-training, CVPR'25

[6] InternVL: Scaling up Vision Foundation Models and Aligning for Generic Visual-Linguistic Tasks, CVPR'24

[7] FILIP: Fine-Grained Interactive Language-Image Pretraining

**Questions:**

Q1. In the text-agnostic contrastive learning part, many works are proposing non-contrastive methods in order to overcome the limitations of contrastive language-image pretraining [1]. Are there any novel aspects that could highlight multimodal alignments between the visual/text individual tokens?

[1] Non-Contrastive Learning Meets Language-Image Pre-Training, CVPR'23

---

### Meta-Review · Area_Chair_GCDa · 2026-01-02

**Summary:**

The average score of this paper is 2 (Min: 2, Max: 2), the concerns raised by the reviewers include the limited novelty, presentation, and experiments. The authors replied to none of the reviewers. After reading the paper and reviewing comments, the AC agrees with the reviewers and tends to reject this paper

**Reviewer Scores:**

NA

---

### Decision · Program_Chairs · 2026-01-26

Reject